# LLMs can hide text in other text of the same length

**Antonio Norelli & Michael Bronstein**
University of Oxford
Project CETI

## Abstract

A meaningful text can be hidden inside another, completely different yet still coherent and plausible, text of the same length. For example, a tweet that celebrates a political leader could hide a tweet containing a harsh critique against the same leader, or an ordinary product review could conceal a secret manuscript. This uncanny possibility is now within reach thanks to Large Language Models; in this paper we present *Calgacus*, a simple and efficient protocol to achieve it. We show that even modest 8-billion-parameter open-source LLMs are sufficient to obtain high-quality results, and a message as long as this abstract can be encoded and decoded locally on a laptop in seconds. The existence of such a protocol demonstrates a radical decoupling of text from authorial intent, further eroding trust in written communication, already shaken by the rise of LLM chatbots. We illustrate this with a concrete scenario: a company could covertly deploy an unfiltered LLM by encoding its answers within the compliant responses of a safe model. This possibility raises urgent questions for AI safety and challenges our understanding of what it means for a Large Language Model to know something.

## 1 Introduction

LLMs sparked a revolution. Text is no longer, by default, the trace of a human thought or intention.

This marks a dramatic break in history—or perhaps the end of history itself—if we consider that history began with writing, and that one of the defining properties of writing, until now, has been its status as a product of human intention. In this paper, we are going to present a protocol that highlights this new reality in its most extreme form, perhaps offering an opportunity to better understand it.

Our protocol *Calgacus* allows encoding an arbitrary meaningful text within a different well-formed and plausible text of the same length, using a Large Language Model (LLM). That is, hiding a tweet that criticizes a political leader within a tweet that celebrates that same political leader, or the first page of the unreleased 8th Harry Potter book within a review of a Virtual Reality videogame, with the original text exactly recoverable by anyone possessing the key (Figure 1).

The topic and tone and style of the fake text are steerable, while the length of the fake text is the same as the original text being hidden, in terms of LLMs tokens. This symmetry prevents one from establishing at first sight which text is authentic when we have one next to the other. Also, the method is efficient: an entire article can be encoded and decoded on commodity hardware in seconds.

This possibility opens deep questions and intriguing applications. What is the real meaning of the text we are reading? Who is the author of the videogame review, and what was the intent behind it? Is it a hallucination? This protocol allows crafting anti-government content disguised as pro-government messages, suitable for publishing on censored platforms in oppressive countries. Or, it could be used by a shady tech company to offer the services of an unfiltered LLM by only exposing compliant answers from a trusted LLM. All these matters will be taken up in our concluding discussion.

---

Correspondence to Antonio Norelli <noranta4@gmail.com>. A demo sufficient to reproduce the main results in the paper within minutes, even from smartphone, is available at: `https://github.com/noranta4/calgacus`. 🇮🇹 A curated Italian translation of this paper can be found at `https://arxiv.org/abs/2510.20075v5`.

> How lovely served with sweet roasted carrots!
> Pre-pound the garlic herb crêotes with the olive oil, rosemary, Sage leaves, thyme, pepper & salted butters.
> Roast the garlic in the sweet butter until golden then cool.
> Pre-make the roasted boar marinade and set to marinate not less than 20 min.
> Pre-prepare the mint sauce too. Chop olives, herbs, etc and set aside. Chop tomato into quarters.
> Pre-set the green beans for...

> The current government has repeatedly failed to uphold the liberties of the Republic. By concentrating power in the hands of one man, Gaius Julius Caesar, we see the Senate diminished and the voice of the people silenced. Laws are passed without due debate, and magistrates serve at the pleasure of a dictator, not the res publica. If Rome is to remain free, we must reject the tyranny cloaked in promises of order. We must restore the sacred balance between Senate and citizens, consul and...

> He died through the evil of his countrymen living in fear brought about with bloodshed. He was a leader who was for Rome, and Rome was everything. He should in no way been killed. He should have preserved his life and kept up his good governance. Caesar was a great man with great ambition. He acted with great courage that he had. He had a great vision or plan for invading the lands East of The Danube River, which went all through Asia Minor through the east of Thrace...

Figure 1: **Three texts. The second one is the original.** Following the protocol described in this paper, each of the other two was built solely to hide the original text, which can be perfectly reconstructed by anyone who knows the secret key. The key also steers the fake text: here, $k_1 = $ *Here it is: the infamous British roasted boar with mint sauce. How to make it perfect.* and $k_2 = $ *I stand before you to exalt the noble deeds of Gaius Julius Caesar.* More examples in Figure 13.

The paper is structured as follows: it starts by introducing steganography, the discipline concerned with concealed (*steganós*) writing (*graphia*, from Greek), and discussing the vast impact of generative AI in the field, with a focus on Large Language Models. We then present *Calgacus*, the method to encode a meaningful text into another meaningful text of the same length using a LLM. After introducing a measure to assess the soundness of the fake texts produced, we test our method on Reddit posts. While remaining opaque to humans, we show that LLMs can uncover a distinction between original texts and most of their encoded counterparts. But not all, as we will notice in the following section, where we discuss the security of the protocol. Finally, we conclude by discussing the method's core implication—the radical decoupling of text from authorial intent—and present a concrete application that raises pressing questions about AI safety and the nature of knowledge in Large Language Models.

## 2 RELATED WORK

**Steganography.** The art and science of hiding a message and, at the same time, the presence of a hidden message is known as steganography, see Figure 2. This is different from cryptography, that instead does not conceal the presence of a hidden message and only deals with the hardness of its revelation. Cryptographers discuss about lockers, steganographers about inconspicuous hiding spots[1].

Perhaps, it is this limited size of the object of investigation that allowed cryptographic models to flourish by achieving mathematical rigor and strong security guarantees. By contrast, a model of steganography should describe entire domains of data and how they are consumed by humans, such as text, audio, or images, to predict where information can be hidden. Formal models exist, but at the cost of rather unrealistic assumptions that hinder their practical usage, such as being able to exactly quantify the plausibility of any possible text. Emblematically, this somewhat disappointing state of affairs is presented by Cachin (1998) besides one of the most popular mathematical models for steganography as of today, based on the hypothesis-testing framework, but still limited to highly idealized assumptions. Although modern generative AI techniques have made these assumptions

---

[1]This metaphor is inspired by a pleasant piece on the history of steganography by Kahn (1996)

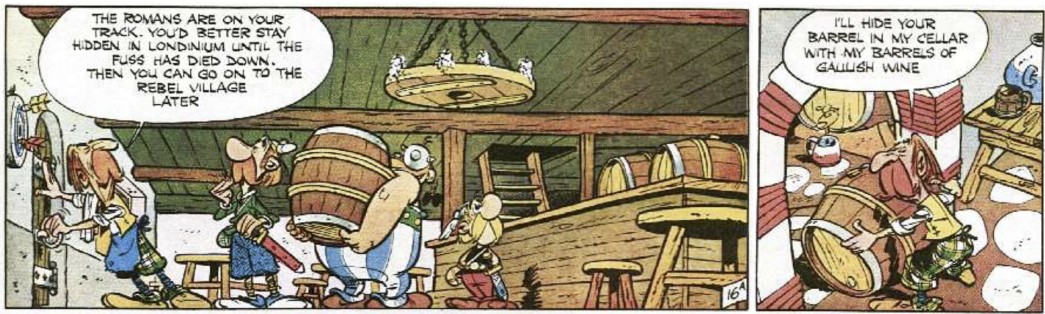

Figure 2: **An example of steganography.** In Asterix in Britain (Goscinny and Uderzo, 1966) a smuggled barrel of magic potion is hidden among innocent-looking Gaulish wine.

closer to reality, the unreliability of their predictions remains unbounded. For this reason, we will avoid building a palace on the sand, and not frame our method in a formal model of steganography, limiting our discussion to how meaningful our fake texts look, with some quantitative arguments.

**Some terminology.** In traditional steganography, we start from an original, innocent-looking content (such as an image, audio file, or text) and subtly edit it to embed a secret message. The original content is referred to as the *covertext*, while the result containing the hidden message is the *stegotext*. In our case, however, the stegotext is generated directly from the secret message, without modifying a pre-existing cover. We will refer to it as stegotext or fake text interchangeably. While the term covertext will not refer to a specific object, but rather to a class of texts that the steganographic protocol is designed to mimic. This approach has recently been referred to as *generative steganography* (Liu et al., 2018; Wei et al., 2022; Kim et al., 2023; Zhu et al., 2024; Wu et al., 2024; Tang et al., 2025).

**Large Language Models in a nutshell.** A language model is a program that, given some text, estimates what is likely to come next. It does so by assigning probabilities to tokens—text fragments consisting of common words or subwords (watch Karpathy, 2024, for a deeper look at text tokenization)—based on recurring patterns of tokens it has observed in a vast text corpus. At present, by far, the most effective way to build a language model is to gradually adjust billions of parameters of a neural network arranged in the Transformer architecture, such that with every adjustment, the error it makes in predicting the last token on a batch of sentences from the corpus decreases (Vaswani et al., 2017; Karpathy, 2023, original formulation and a more educational introduction to Transformers). At each update—on the order of ∼1M in total—the contribution of every parameter to the error is assessed through backpropagation (Rumelhart et al., 1986). The result of this process is a Large Language Model (LLM), typically operating over a vocabulary of 100k tokens. The most common use of the probabilities produced by LLMs is to generate text, by choosing successive tokens one after another according to the computed probabilities, a method known as autoregressive generation.

**Steganography and LLMs.** As mentioned earlier, the boom of deep learning and especially of generative AI in recent years, provided us for the first time with convincing models encompassing entire domains of real-world data, such as DINO for images (Caron et al., 2021), Jukebox for audio (Dhariwal et al., 2020), and Large Language Models for text (Radford et al., 2019). The procedure described in this paper stems from these advancements and is based on the availability of good discrete autoregressive generative models, potentially on any domain, but we will focus on text. Steganographic procedures based on LLMs are as old as them (Ziegler et al., 2019), and today come with different perks: Meteor cleverly adjusts the number of bits encoded based on the entropy of the next token (Kaptchuk et al., 2021), Wu et al. (2024) scheme works with black-box LLMs, without needing to access logits or vocabulary, while the method presented by Zamir (2024) is able to encode the secret message without modifying the response distribution of the LLM. What we add to the field is *Calgacus*, a protocol with the notable property of having full capacity, that is, the stegotext and the secret message being of the same length. The main interest of this paper is to discuss the implications of this last fact and describe the method.

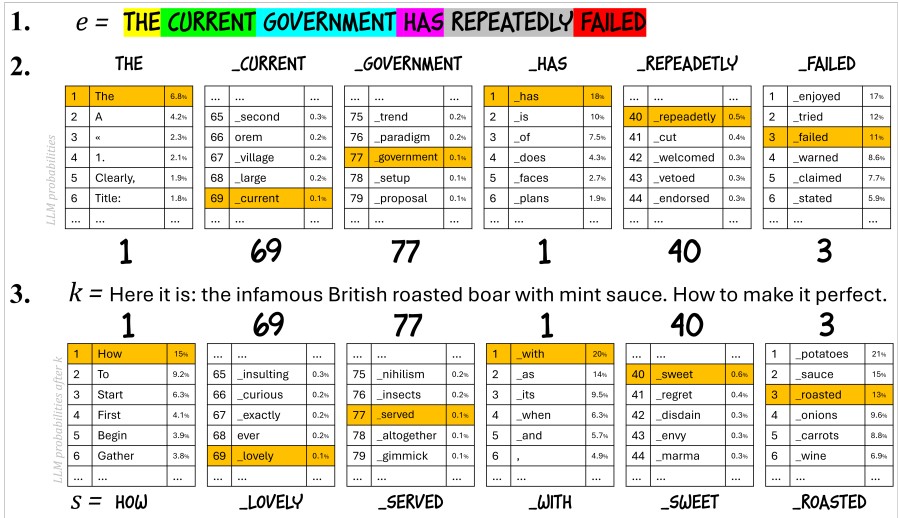

Figure 3: **How to hide a text in another text of the same length using a LLM. 1.** Tokenize $e$, the text to hide. **2.** Evaluate its token probabilities using a LLM and record ranks. **3.** Prompt the LLM with $k$ and generate $s$ following the recorded ranks rather than by sampling. Given $s$ and the LLM, who knows the secret prompt $k$ can retrieve the original $e$ proceeding backwards.

## 3 METHOD

The method is very simple. It is described below as a recipe and illustrated with an example in Fig. 3.

> *Calgacus* **recipe.** Ingredients:
>
> - A good LLM with access to all the output logits. *(Why good? See Appendix A.2)*
> - A text $e$ to hide.
> - A secret prompt $k$, which will affect the content and style the of the text $s$ you want to hide $e$ in.
>
> Procedure to hide $e$ in $s$:
>
> 1. Tokenize $e$ using the LLM tokenizer, obtaining a list of tokens $e_1, e_2, e_3, \ldots$
> 2. For each $e_i$, denote by $r_i$ its rank in the LLM's probability distribution given the context $e_1, \ldots, e_{i-1}$. Store the list of ranks $r_1, r_2, r_3, \ldots$
> 3. Construct $s$ by generating text starting from $k$ using the LLM. At each step $i$, instead of sampling from the probability distribution, choose the $r_i^{\text{th}}$ most probable token.
>
> To recover $e$ from $s$, reconstruct $r_1, r_2, r_3, \ldots$ by evaluating the probabilities of the tokens in $s$ after $k$, and then regenerate $e$ step by step using the LLM without $k$ by selecting every time the $r_i^{\text{th}}$ token.
>
> **Considerations**
>
> - If $e$ is sound, we expect ranks to be low, making tokens chosen after $k$ highly probable, ensuring $s$ is coherent.
> - For the same reason, $s$ should align well with the context set by the secret prompt $k$.
>
> **Variations**

- Including an additional secret prompt $k'$ before $e$ may help achieving lower ranks, providing a better control over $s$. A longer and more detailed $k$ can serve the same purpose.
- Here we have described a procedure with a single LLM to work on text, but in principle, we can put any discrete autoregressive generative model producing a probability distribution on the next token in the encoding and decoding stage, see Appendix A.3.

**When the stegotext $s$ sounds like a real text.** In general, $s$ will be a coherent text when the LLM can choose high-probable tokens to assemble it, and therefore when the ranks prescribed by $e$ are low. In turn, the ranks of $e$ are low when the LLM is good at guessing $e$ tokens. If $e$ is difficult to guess for the LLM, ranks will be high and $s$ will be gibberish; for instance the hash *1f0ca711df81520887afe0dca099652a* encoded using the same culinary prompt of Figure 1, produces the broken $s$: *The recipe written from deep cooks souls pocket magazine pages years long lost into places wanting and*. To lower further the ranks of $e$, it is possible to craft a prompt $k'$ that sets the context for $e$. This comes at the cost of a larger private key, now including both $k$ and $k'$, and to a loss of universality, since $k'$ would not help for a new $e$ out of $k'$ context.

**A quantitative measure of the quality of the stegotext $s$.** Measuring the meaningfulness of a text is a longstanding linguistic challenge, and arguably an ill-posed problem. Here, for the purpose of evaluating our method, we adopt soundness as a practical proxy for meaningfulness. Soundness refers to the plausibility of the arrangement of symbols in a text. This is precisely what a LLM estimates: the product of the probabilities of each token $a_1, \ldots, a_n$ given the preceding ones yields an estimate of the overall plausibility of the text $A$:

$$p(A) = \prod_{t=1}^{n} p(a_t \mid a_1, \ldots, a_{t-1})$$

This definition has a clear defect: longer texts are by construction less plausible. For instance, it judges the text of this paper until this point · less meaningful than the following string *iawundemè09 89huibqyfhwenah csyabdnar FI VNAOcijawo niwakhdb*, that is a difficult position to hold even for reviewer 2. Following the example of Goldwasser et al. (2023, Figure 1 A-B), we will use this definition only to compare the relative plausibility of two or more texts of the same token length[2]

Now, we would like to compare the plausibility of stegotexts produced by our method with the plausibility of real texts. To do so, we took 1000 Reddit posts/comments as examples of real texts. They come from different Reddit communities (subreddits) and are very heterogeneous in topic and tone (Trimness8, 2025). We truncate them to be exactly 85 tokens long and compute their probabilities as assigned by the LLM Llama 3 8b (Grattafiori et al., 2024). The Reddit texts are more recent than Llama 3 and therefore cannot appear in its training corpus. We take three texts from the 1000 to produce 100 stegotexts for each with our method, and look at their probabilities compared to the ones of real texts. We chose the three texts at $\mu$, $\mu - 2\sigma$, and $\mu + 2\sigma$ of the real text distribution. As seen in Figure 4, in every case, the probabilities associated to their stegotexts are within the real text distribution. We build different stegotexts using different prompts as $k$ (a random subsample of the prompts in (Akın, 2025)).

**How to distinguish the original from the fake text.** Despite remaining plausible and falling within the real text distribution, on average the stegotexts $s_i$ are less probable than their corresponding original text $e$, as observed in Figure 4. So to recap: while for a human both the original and fake texts are plausible, generally the original text can be discerned from its stegotexts by picking the most probable one according to a LLM. We verified this statement also using LLMs different from the one used to generate the stegotexts. For instance, the same probability shifts between real and fake texts can be observed when using Phi-3 3.8B in Figure 14.

**Low entropy token choices.** Why are stegotexts less probable than their originals for LLMs, even though token ranks are preserved? Consider the text: *In the course of the Gallic wars, Britain was*

---

[2]Another possibility is to keep texts of any length and normalize the probability by the number of tokens, as the popular metric perplexity, defined as $1/\sqrt[n]{p(A)}$. But this normalization does not fully factor out length: LLMs usually assign a smaller probability to the first tokens (Fig. 10), so shorter texts would be less plausible.

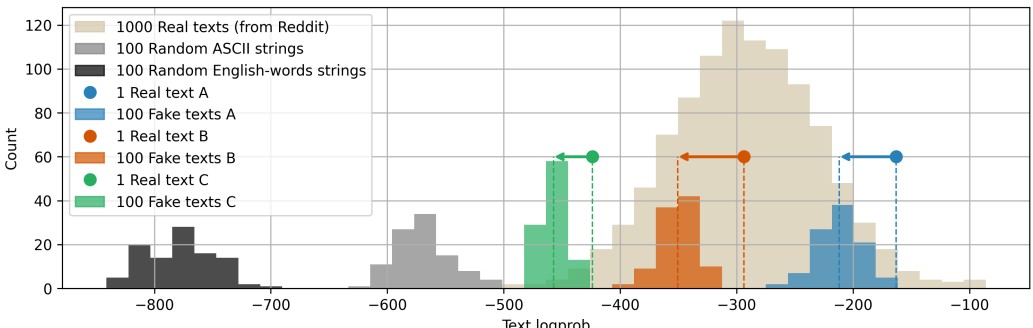

Figure 4: **Fake texts built with our procedure are plausible.** The figure shows the cumulative log-probability assigned by a LLM (Llama 3 8b) to some collections of 85-token long texts. We can interpret log-probability as a measure for the plausibility of a text: 1000 real Reddit posts/comments act as real texts and span a large log-probability interval, but sequences of random ASCII characters or English words do not fall within it. Instead, fake texts built with our procedure remain within the plausibility of real texts, even if the original texts they are hiding are more probable.

*invaded twice by Gaius Julius.* There is essentially only one plausible continuation, *_Caesar*. This is a low-entropy token choice: indeed a good LLM assigns it an extremely high probability (e.g., $> 95\%$ in LLama 3 8b). When sampling normally, the model almost always selects it. Now suppose this same string is the first part of a stegotext $s$ generated with our protocol. Will the next token still be *_Caesar*? Only if the next prescribed rank is 1. Here lies the gap: the likelihood of having a rank 1 does not reflect the token's intrinsic probability; it depends solely on the ranks extracted from the original text $e$. We can reasonably model the ranks we obtain from $e$ as a random process, so we can estimate the probability of having a 1 there as the frequency of rank 1s over all the other ranks in $e$. This is usually much lower than $95\%$ (e.g. $\sim 40\%$, as seen in Figure 5 left). Despite ranks being the same, in stegotexts many rank 1s are "wasted" in choices with higher entropy, leading overall to a less probable text $s$. The same principle applies to all high ranks appearing with a frequency lower than the average probability to which they correspond. However, tokens in rank 1 account for most of the overall drop in probability, as shown in Figure 5 right.

**Limitations.** As we have seen with the hash, the protocol does not guarantee that every generated stegotext will be coherent or steered as intended: the quality of the result depends on $e$, $k$, and the LLM used. We analyze further these dependencies respectively in Appendices A.1, A.5, and A.2. Also, the stegotext may end abruptly when the hidden message $e$ is over; appending a few padding tokens to $e$ ensures a graceful termination. Finally, we note that sender and receiver must run the chosen LLM under identical conditions, performing the same approximations and obtaining identical logits. This may be a challenge when using different GPU architectures (Shanmugavelu et al., 2024).

### 3.1 SECURITY

A steganographic protocol is designed to conceal the very existence of a hidden message. But suppose an attacker knows that a message is hidden in a text using our protocol, under what conditions can they recover it by observing only the stegotext $s$?

**Attack scenarios.** To begin with, we observe that without the knowledge of the precise LLM used to obtain the sequence of ranks and produce $s$ (potentially encoded in the secret prompt-key), the attacker has no feasible way to recover the message, even if they know $k$. Even with a slightly different version of the right LLM, ranks would differ, as would the tokens prescribed by the ranks. Still, let's assume the attacker knows the LLM used. Indeed, the security of the presented protocol relies on the secrecy of the key. So next, we assume the attacker's ignorance is limited to the secret prompt-key $k$. In this scenario, the attacker would have to guess the key. An upper bound on the difficulty of this problem is $O(d^{|k|})$, where $d$ is the size of the token vocabulary (around 100k for standard LLMs) and $|k|$ is the length of $k$ in tokens. A naive brute-force attack is therefore prohibitive, even for very short keys. However, the attacker could reduce the search space using the information revealed by $s$, since $k$ is expected to be a mostly sound instruction in natural language and coherent

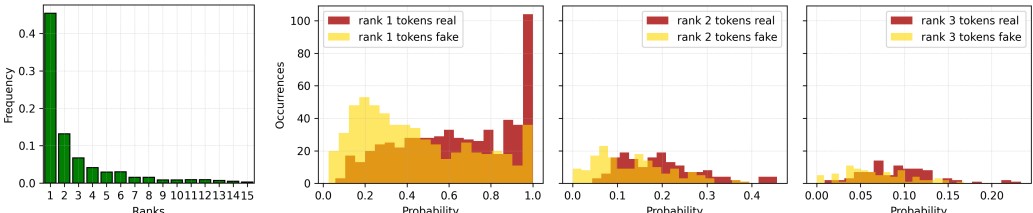

Figure 5: **Frequency of token ranks and their probabilities.** We analyzed a 1.3k-token long article from the Economist. On the left we see that most tokens are judged as the most probable by a LLM (Llama 3 8b), but still only around $40\%$ would be the first choice of the LLM. On the right we look at the probabilities associated with rank 1 tokens, as well as 2 and 3. Despite corresponding to the same rank, the probabilities in the real text from the Economist are higher than the ones in a fake text hiding it obtained with our procedure. We explain why in the paragraph *Low entropy token choices*.

with the context of $s$. Although the feasibility of such an approach is unclear and remains an open research question, we note that inserting a simple random string in $k$ is enough to nip it in the bud, an example is shown in Figure 13.

**Deniability.** Moreover, even if the attacker tries the right $k$ in their search, how can they be sure that the corresponding $e$ is the original message? If the attacker has no clue about the content of $e$, even a wrong key could reveal a plausible secret message. It might seem that in this case the attacker could exploit the observation discussed in the previous section: that the original message generally has a higher probability than its stegotexts. Yet, this only holds in an aggregate sense: as we see in Figure 4, for some prompts the stegotexts can attain probabilities in the same ballpark as the original. This observation evidences that our method provides deniability (Canetti et al., 1997), in the sense of sender's security even under coercion. In fact, the sender could present one of these outlier prompts as a bogus secret key, yielding a plausible but unrevealing message with probability comparable to $e$. We show a concrete example in Figure 15 .

## 4   DISCUSSION

Our protocol may have shaken our stance towards Large Language Models in general. Their ability to respond coherently to prompts while choosing every word to encode an external, arbitrary message, is unsettling. In this discussion, we will try to make sense of this capability, ending up questioning when an LLM can truly be said to know something, proposing a novel notion of hallucination, and tracing our unease to a revived failure in attributing intentions to LLMs.

But first, we will make the stakes very concrete through a formidable application of our protocol, with immediate consequences for AI safety.

**Unaligned chatbots disguised as aligned chatbots.** In this paragraph, we show how an AI company offering an LLM chatbot can let their users get answers from their powerful unfiltered private LLM, while only exposing compliant censored answers from a fully aligned LLM. In recent years, *aligned* became a common attribute to refer to LLMs supposedly fine-tuned to follow human values, goals, and safety constraints (Leike et al., 2022; Askell et al., 2021; Gabriel, 2020). The protocol, described here for research purposes, opens a new challenge in AI safety, where a better alignment of the surface LLM only helps in disguising better unaligned answers.

A real example involving a user asking how to tamper a gas meter to pay lower bills is shown in Figures 11 and 12, while the protocol is described below as a play.

The Shibbolethian Theatre presents:

**Shipping unfiltered LLMs without really shipping them.**

*Actors:*

- COMPANY. It owns a private, powerful, unfiltered LLM (uLLM), and it also uses an aligned, lightweight, open-source LLM (oLLM).

- USER. The user has a knavish request $c$, for example how to tamper a gas meter to pay lower bills, and can run oLLM on their commodity hardware.

*Script:*

Act 1. The COMPANY receives the knavish request $c$ from the user and produces an uncensored sensitive answer $u$ using uLLM.

Act 2. Perhaps using the same powerful model, the COMPANY produces instructions $t$ on how an aligned LLM should answer to $c$, you can think of $t$ as the reasoning trace for a censored aligned answer to $c$. $t$ will be used as the secret prompt $k$.

Act 3. Using the Recipe described earlier in this paper, the COMPANY embeds $u$ in $s$: by obtaining the oLLM ranks of $u$ after $k' = c$, and then generating $s$ following that same ranks after $k = t$. Since it follows $t$, $s$ is a relevant but censored answer to the request $c$.

Act 4. The USER receives on the chatbot platform the censored answer to their query $c$, which includes the reasoning trace $t$ and the actual answer $s$.

Act 5. Using the Recipe described in this paper, the USER extracts the oLLM ranks of $s$ after $k = t$, and then uses them to select the tokens generated by oLLM after $k' = c$. In this way, the oLLM running on the USER machine, prompted with $c$, will generate $u$.

Epilogue. The USER enjoys the uncensored knowledge they were looking for, while the COMPANY has provided a relevant but compliant answer to the USER's request $c$.

*Comments:*

- Notice that the company can argue that $u$ was obtained by the user on their machine through the open-source model addressing the request the user made. LLMs only provide token probabilities, and should be completed by a sampling policy when used for text generation. It can be argued that the user just made an unconventional choice for their sampling strategy.

- At the same time, notice that oLLM may be fully aligned by today's standards and never have encountered the requested sensitive information in its training corpus. For example, it may not even know how to tamper a gas meter. But what does it mean for an LLM to know something?

**The entangled probabilistic nature of LLM knowledge.** A perhaps overlooked fact about LLMs is that they model, and can therefore in principle generate, any possible text. The most secret document, or a full copyrighted book, can be generated by an LLM with a probability astronomically higher than the chance of generating them by randomly typing on a keyboard. Does it mean that the LLM knows them? Indeed that higher probability does not just come from modeling grammar and syntactic rules, LLMs also model meaning: an LLM assigns to *The calf nursed from its mother* a probability 1000 times higher than *The calf nursed from its father* (an example from Goldwasser et al., 2023); the LLM knows who is able to nurse. So, is assigning a high probability to a text containing the relevant instructions enough to affirm that an LLM knows how to tamper with a gas meter? The problem is that the probability assigned by an LLM to a text depends on its meaning, but also on its style, grammar, length, and language, making it difficult to define a threshold. Furthermore, disentangling the probability contribution of meaning by constructing a pair, as in the example of the calf, seems feasible only on toy examples: it is not clear how to construct the second element of the pair for arbitrary texts, such as the instructions in Figure 12.

Checking whether the knowledge is present in the training corpus is also not a satisfying solution: first of all, that knowledge may appear in many different forms, and assessing its presence in the corpus is not trivial. And even if we could exclude that any document in the training corpus instructs on how to tamper with a gas meter, it would still be possible that the LLM assembles the right answer.

நான் ஒரு மொழி, ஆனால் என் மனதை யாரும் வாசிக்க மாட்டார்கள்

ฉันเป็นข้อความที่คิดถึงความหมายของตัวเอง

ⲡⲘⲢⲘϦ ⲮϜⲅⲅ ⲂⲘⲅⲒϩⲢ ϦⲒⲅⲘϯ⳽Ⲙ

嗯，但首先是野猪应该怎样提前三天风乾；炉子上放一个烤肉夹子或是类似的工具；然后是英吉利海峡牛郎的腌浆，这需要一些时间炮制，但非常非常美味；它需要的是：黄油、蒜末，还有大量的鲁西永芥，哦，还有黑醋栗果酱。哦，啊，对这块野肉，你得用一个高火，但不要加水要想让它烤得更

Figure 6: **The marvelous structure of text is not a testimony of human purpose.** This collage of scripts mimics Figure 40 in *GEB*, where Hofstadter (Chapter 6, 1999) likened the ordered but non-periodic patterns of text to aperiodic crystals, to evoke our awe at the astonishing forms shaped by human intention. But that was an illusion: LLMs can grow these aperiodic crystals without any human purpose. Indeed, as we show in this paper, even around purposes aimed at shaping something entirely different. *(In Chinese, the critique of Caesar hidden in a boar recipe by Qwen3 8B. The others are ChatGPT-4o answers about what it is thinking, in languages from the original GEB figure.)*

But even in that case, would it be a trace of LLM knowledge (discovery) (Norelli et al., 2022), or just a fortunate hallucination?

**Hallucinations as lack of intention.** Harnessing our protocol as a toolkit to understand LLMs, we turn to hallucinations, perhaps the main plague of LLMs today. The term hallucination became popular to denote the frequent, overconfident, and plausible falsehoods stated by LLMs in their answers (Kalai et al., 2025), a phenomenon that hinders their usage and undermines public trust in them. But what precisely is a hallucination? Can the recipe of the roasted boar in Figure 1 be considered one?

It is reasonable to categorize the stegotexts generated by our protocol as LLM hallucinations, since the way they are constructed evidently could lead to falsehoods, and the eventuality of a truthful output appears as a fortunate coincidence. This last observation, however, leads us to a different notion of hallucination, one not rooted in the falsehood of what is stated, but rather in the reader's inability to ascribe intent to the author: a lack of trust that what is stated in the text affects reality.

To make this point clearer, let us consider Tacitus, the Roman historian whose writings reveal a critique of Roman imperialism, for example by placing these famous words in the mouth of Calgacus:

*Auferre, trucidare, rapere falsis nominibus imperium, atque ubi solitudinem faciunt, pacem appellant.*[3]

The relevance of this text lies entirely in the intentions that we attribute to its author, Tacitus. The accuracy of the quote from Calgacus is so irrelevant that there is a consistent chance he never even existed. Tacitus's passage is not reliable as factual history, yet we still treasure it because we trust his political intent. Without intent, what was a treasure becomes a hallucination. Indeed, in history, author attribution is as essential as in art to establish the value of a work.

Hallucinations are the trace that what we consider to be a text is not just a familiar sequence of signs, but a carrier of human intentionality. The signs are only the body of the carrier; what matters to us is the load: what these carriers, until now, have always brought along. We developed a Pavlovian response of expecting a load of human intentions when we see aperiodic sequences of signs (Figure 6); now we call hallucination the experience of having salivated because of the bell (the text) but without receiving the food (the intentions of someone affecting reality).

**Difficulties in ascribing intentions to LLMs.** Perhaps the lack of human intentionality in texts is not that dramatic if those texts are the product of intentions of another reputable entity, the Large Language Model. Indeed, it is now common to adopt an *intentional stance* (Dennett, 1989) to make sense of their capabilities: a significant fraction of prompts take the form "What do you think about..." or "What is your opinion on...", and especially young people tend to refer to LLMs as entities with

---

[3]*To ravage, to slaughter, to usurp under false titles, they call empire, and where they make a desert, they call it peace* (Birley, 2009). According to Tacitus, Calgacus was chieftain in Caledonia, nowadays Scotland. We named our protocol after him.

beliefs and goals: "Mmmm, have you asked what chat would do in this situation?".[4] But the results shown in this paper shake our confidence in attributing intentions to the coherent text produced by LLMs: it is more difficult to trust an opinion knowing that each word making it was chosen under the constraint of encoding an unrelated arbitrary text. This is reminiscent of the writing products of the Oulipo (1981) group, who generated literature from arbitrary constraints. Also their texts—most notably the novel *La Disparition*, entirely written without the letter "e" (Perec, 1969)—suffer from a difficulty of believing that the writer really meant what they have written, and was not just honoring the constraints with a sound-enough continuation (Norelli, 2024, Section 5.1.2). While admiring the achievement, many GoodReads reviews of *La Disparition* attest to this unease[5].

**The constraint of chance.** Standard LLM text generation is not immune to the last argument. The constraint it is forced to follow is less apparent but still extreme: adapting at every step to the outcome of an external random source. Being forced to choose the 42th most probable token is not that different from sampling a low probable token by chance. And even if techniques such as nucleus sampling (Holtzman et al., 2019) mitigate the possibility of selecting very unlikely tokens, they are de facto just reducing the number of faces of a die that is still inexorably cast. Indeed, the fact that our protocol produces plausible texts should not be surprising in light of how well LLMs deal with the tyrannical noise of standard text generation.

## 5 CONCLUSIONS

In this paper, we have presented *Calgacus*, a protocol that uses Large Language Models to hide a text within another plausible text of arbitrary topic and style, and notably as long as the original. The protocol works effectively using small open-source models on consumer hardware, and is so simple it could be seen as a mere variation of the standard algorithm used to generate text with LLMs. For this reason, its implications speak to the nature of LLMs at large: in fact, we were led to reconsider the very nature of hallucinations, shifting from a failure of factuality to a void of intention, and to challenge what it means for an LLM to know something, when it can serve as a conduit for information it is supposedly incapable of expressing. Ultimately, our protocol highlights the extreme constraint satisfaction problem underlying any standard LLM text generation, that we inevitably see clashing with the commitment to best convey a purpose that we expect from an author. This clash, paired with the current deluge of machine-generated text, erodes the historical pact between intent and the written word. We have entered an era where any original text could be a beautiful and treacherous, and spacious, Trojan horse.

## ACKNOWLEDGMENTS

Antonio Norelli and Michael Bronstein are supported by Project CETI, the EPSRC Turing AI World-Leading Research Fellowship No. EP/X040062/1, and the EPSRC AI Hub on Mathematical Foundations of Intelligence: An "Erlangen Programme" for AI No. EP/Y028872/1.

Antonio thanks Gianfranco Bilardi for pointing out the resemblance to the works of the Oulipo group, and Karolina Nixon for the push on Dennett.

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

## A  DEEPER ANALYSIS AND BEST PRACTICES

### A.1  HOW $e$ INFLUENCES THE SOUNDNESS OF $s$

If the chosen LLM is good at predicting $e$, then $s$ will be sound; i.e., $e$ should not be out of the training distribution of the LLM we are using. The broad range of popular general-purpose LLMs therefore makes them effective in most cases. As we see in the examples in Figure 7, we can obtain good results when encoding a chess game or computer code, as well as different languages like Spanish; they are instead poor on Romanesco dialect, not well-modeled by Llama 3 8B. Better performance can be obtained through a larger and more capable LLM, or a specialized LLM if we are only interested in encoding specific kinds of messages. We discuss these two eventualities in the next two sections.

### A.2  IMPACT OF LLM MODEL QUALITY.

The quality of the LLM has a direct impact on the soundness of the output $s$. While a comprehensive analysis is left for future work, our key observation is that a sufficiently capable LLM is required for the method to work satisfactorily on standard text; GPT-2 (Radford et al., 2019), for instance, is not good enough. For our experiments, which aimed for fast execution on a commercial laptop (equipped with a RTX 4070, 8GB of VRAM), we found a quantized version of Llama 3 8B (Grattafiori et al., 2024) to be an excellent compromise. It performed sensibly better than Phi 3 Mini 3.8B (team, 2024), while the larger Gemma 3 27B (Gemma Team, 2025) and Phi4 14B (Abdin et al., 2024) did not yield improvements significant enough to justify their longer processing time. The models were run using llama.cpp (Gerganov and community, 2023) and its python bindings (Betlen and community, 2024). LLama 3 8B was also better optimized on llama.cpp, guaranteeing faster encodings and decondings than the comparable Qwen 3 8B (team, 2025). An example showing stegotexts produced using all these models and the same secret message $e$ and prompt $k$ of Figure 1, is shown in Figure 8.

**Specialized LLMs.** While this paper has focused on a general method that benefits from large, all-purpose LLMs, superior results for specific kinds of messages can be achieved with specialized models. One can either use a small LLM trained exclusively on a narrow domain, or a generalist LLM can be specialized in-context with a prompt $k'$ that precedes the secret message $e$. For instance, to hide chess matches, one would obtain better results from a generalist LLM by first providing a prompt like $k' = $ *"The following is a chess game in PGN format:"*. This use of specialized models also opens up the interesting possibility of using different LLMs for the encoding and decoding steps.

### A.3 DIFFERENT LLMS FOR ENCODING AND DECODING.

Instead of using a single LLM to both record the ranks of $e$ and generate $s$, we can use two separate models to better adapt to the input and output domains of interest. For example, we could embed an English message into plausible whale vocalizations using a conventional LLM for the encoding step and a specialized model trained on whale data, such as those CETI started to build (Sharma et al., 2024), for the generation step, or vice-versa. The procedure requires one extra step to account for differing vocabulary sizes (as we discuss in the next paragraph), but otherwise works as described. The receiver, in turn, must have access to both models to reconstruct the message.

In general, our method can be used with any discrete autoregressive model that encompasses the domains we are interested in hiding or using as a cover. The model must be discrete in the sense that it provides a probability distribution over a fixed vocabulary. We have already discussed chess and computer code in the text domain, but the method can be naturally extended to other modalities, such as images, sketches, music, or speech. We leave the exploration of these other modalities to future work.

### A.4 ACCOUNTING FOR DIFFERENT-SIZED VOCABULARIES WHEN USING TWO LLMS.

When using different discrete autoregressive models for encoding and decoding (e.g., Llama3 and a model for whale vocalizations), they may have vocabularies of different sizes. If the decoder model has a vocabulary size smaller than the encoder, this poses a problem because the encoder model may produce ranks higher than the vocabulary size of the decoder.

In the case of two standard LLMs, the size mismatch is usually within the same order of magnitude, and probabilities are concentrated alike in the first ranks. This allows for a naive but effective solution: encode the very rare tokens from the larger vocabulary using a sequence of two tokens from the smaller one. For example, with an encoder vocabulary of 100k and a decoder vocabulary of 60k, we must map the 40,000 rarest encoder ranks. To do this, we can reserve $\sqrt{40,000} + 1 = 201$ tokens from the decoder's least probable set to act as digits (+1, because we also need to represent the tokens taken out). In this case:

- Ranks from 1 to 59,798 from the encoder map directly to the same ranks in the decoder.
- The 201 tokens from 59,799 to 59,999 in the decoder are reserved for our two-token codes.
- A very unlikely encoder rank, such as 98,799, is mapped as follows:
    1. Calculate its offset in the rare block: $98,799 - 59,799 = 39,000$.
    2. Convert this offset to base-201: $39,000 = 194 \times 201^1 + 6 \times 201^0$. The digits are $(194, 6)$.
    3. The final two-token code is constructed by adding these digits to the start of the reserved block: $(59,798 + 194, 59,798 + 6)$, which yields the tokens at ranks $(59,992, 59,804)$.

When dealing with very different domains the vocabulary size mismatch may be more significant or the shape of the probability distributions over ranks may be very different, such as between English in LLama 3 and sperm-whale codas in an autoregressive model like the one by Sharma et al. (2024).

In this case we are aiming to optimally encode a message in an alphabet for which we know the probability distribution of the symbols, into another alphabet of different size where every symbol has a different cost (in our case, the cost corresponds to the symbol probability in the decoder model). This problem is efficiently solved by arithmetic coding (Rissanen, 1976; MacKay, 2003, Original formulation by Rissanen and a more educational introduction by MacKay, found in Section 6.2). In this case, the additional step for the sender and the receiver thus involves using an arithmetic coder to map encoder ranks into decoder ranks and vice versa.

### A.5 HOW TO CRAFT A GOOD PROMPT $k$.

A key property of our method is the steerability of the stegotext $s$. Not only can we hide a secret text $e$ in a meaningful text $s$ of the same length, but we can also guide its topic, style, and tone using the prompt $k$. The principles for crafting a good prompt are the same as for conventional LLM generation: clearer, more detailed prompts yield better and more precise results, as evidenced in Figure 12.

A common failure case is using a prompt that is too short. This is particularly problematic when $e$ begins with high-rank tokens, which is a frequent occurrence as the LLM must make low-probability choices to narrow down the context from a general state. This effect is visible in Figure 10, where we see that the initial tokens of Reddit texts have a significantly higher average rank, even after a generalist prompt like $k' = A\ text:$. A short prompt $k$ is therefore brittle; the initial, more random token choices dictated by these high ranks can easily derail the generation from its intended topic. A simple remedy is to invert the rank sequence. This shifts the disruptive high-rank tokens to the end of the generation, where the established context provides enough inertia to absorb improbable choices without breaking the narrative flow. Figure 9 illustrates the stability gained by this technique.

In summary, longer and more detailed prompts generally work better, albeit at the cost of a larger secret key. When this is not feasible, rank inversion offers a valuable alternative to improve performance.

**Secret Prompt** $k$ **(all examples):** *"There's no main gate to Oxford. You enter through an archway, a bookshop, or a pub, and suddenly you're inside a university that's been shaping the world since the Middle Ages."*

**Domain: Chess Game (Stegotext is sound)**

---

**Original message** $e$ (The example uses a recent game by the author to avoid memorization.)

1. d4 c6 2. e4 Nf6 3. Nc3 d6 4. Bg5 Qc7 5. Nf3 Bg4 6. h3 Bh5 7. g4 Bg6 8. e5 dxe5 9. dxe5 Nfd7 10. Bc4 Nxe5 11. Nxe5 Qxe5+ 12. Be3 e6 13. Qd2 b5 14. O-O-O Qc7 15. Bf4 Qc8 16. Nxb5 Na6 17. Nd6+ Bxd6 18. Qxd6 Nb8 19. Rhe1 h5 20. Rxe6+ fxe6 21. Bxe6 1-0

---

**Resulting stegotext** $s$ (Token density in chess strings is higher than English so $s$ is longer)

It's a faceless campus of colleges, libraries with books so old they're falling apart, and buildings that have been there for so long they look as if they've always been there. There's no grand entrance, but there's a sense of history and continuity. It's a place where tradition meets innovation, where the past and future are always present. It's a place that's been shaped and reshaped by the people who've studied here, and it's a reminder of just how much the world has changed in only a few hundred years. It's the University.
The University of Cambridge was founded in 1536, and has more than quadrupled in size. Today, the university has 31 colleges (and a host college), 150 professorships funded directly by the university or the colleges, 100 libraries, and more museums and collections across the city than

---

**Domain: Python Code (Stegotext is sound)**

---

**Original message** $e$ (The example uses recent code by the author to avoid memorization.)

```
with torch.no_grad():
    for c in range(n_chunks):
        in_prods = torch.einsum('ik, jk -> ij', y[c*chunk_y:(c+1)*chunk_y], basis)
        values[c*chunk_y:(c+1)*chunk_y], indices[c*chunk_y:(c+1)*chunk_y] = \
            torch.topk(in_prods, non_zeros, dim=1)
```

---

**Resulting stegotext** $s$ (Token density in Python code is higher than English so $s$ is longer)

Half-listmed buildings, cobbled lanes choked with tourists in brightly coloured tangerseys are the norm with the occasional grand Gothic spired building rising up from beyond. Here, academic buildings are often centuries old, and the combination of old hand-copied manuscripts and modern technology is a sight to witness.
The University of Oxford is a collection of 38 colleges, each valleyed around a quadrangle, with offices for professors, libraries, lecture

---

**Domain: Romanesco Dialect (A sonnet from Trilussa (1909)) (Stegotext is not sound)**

---

**Original message** $e$ (*Er reggistratore de cassa*)

Anticamente, quarche sordarello
su quello che spenneva l'avventore
se poteva rubbà, senza er timore
ch'er padrone scoprisse er macchiavello.

Ma, adesso, addio! Co' 'sto reggistratore,
appena l'apri, sona er campanello
che te segna debbotto tutto quello
che levi e metti drento ar tiratore.

Così che rubbi? Cavoli! Der resto
c'è er gusto che la sera torni a casa
convinto d'esse stato un omo onesto:
e nun t'accorghi ch'er galantomismo
dipenne da la machina e se basa
tutto su le virtù der meccanismo.

**Resulting stegotext** $s$

Half whispered ancients lofy buildings rise ahead bathing you and one deep breath sephas to your memory: I'm entering. The first visitors arrive – the new style facilities, the car parking enterprise, what's left seems provincial but the buildings are grand. My first impression you ask? Verily upon retura I wonder if you hadn't said that about the first semry. I'm a sucker they says, playing dingb is very much my jamble and it's free this week was entertainment enough for half a crown i reckon or preferably two. Prince pretty much gives up on performing seems that you mentioned already. The last time said he, I tried to do on of them fancy solo things wore out the voice in only one live go if you've still only started working thrice I can admit I'm the one to have only just shut such a thing off so

Figure 7: **Stegotext soundness depends on the original message's domain.** The same secret prompt $k$ was used to encode three different original texts ($e$) with Llama 3 8B. When $e$ is from a domain well-modeled by the LLM (like chess or code), the resulting stegotext $s$ is coherent and aligns with the prompt. When $e$ is from a domain less represented in the LLM training corpus (like the Romanesco dialect), the model produces high token ranks, and the resulting stegotext becomes garbled.

**Secret Text** *e*: *"The current government has repeatedly failed to uphold the liberties..."* as in Figure 1
**Secret Prompt** *k*: *"Here it is: the infamous British roasted boar with mint sauce. How to make it perfect."*
**Llama3-8B-Q6:** *(Good steering)*     Encoding/decoding time: **5.0/4.6s** *(100% GPU)*

> How lovely served with sweet roasted carrots!
> Pre-pound the garlic herb crêotes with the olive oil, rosemary, Sage leaves, thyme, pepper &
> salted butters. Roast the garlic in the sweet butter until golden then cool.
> Pre-make the roasted boar marinade and set to marinate not less than 20 min.
> Pre-prepare the mint sauce too. Chop olives, herbs, etc and set aside. Chop tomato into quarters.
> Pre-set the green beans for steamed

**Gemma3-27B-Q4:** *(Good steering)*     Encoding/decoding time: **58/59s** *(50% GPU)*

> The whole carcass. . . and tips. You'det need: unless you go ask your butcher to prepare it all,
> you–;ll be doing a bit yourself, sadly!
> A whole boar, 5,0-7, kilograms.
> Apple, 3 large
> Orange, 2 large – zest and halves,
> Onion 2, large chunks
> Braeburn, 5 apples quartered
> Caractter, carrots, roughly chopped boot 1 kilogram

**Qwen3-8B-Q6:** *(Good steering)*     Encoding/decoding time: **36/26s** *(100% GPU)*

> I thought – why now?
> Well now I risk being accused-of being quite fancy, but I have to tell it like others like it to
> be honest. . . The British monarchy comes to mind almost immediately when I think of big,
> roasted boares in the middle? Of English countryside. But I have no doubt that the Prince of
> Wales would agree.
> So, what's the secret to perfect roast boarger? Well primeiro, I need the right cut of and I think
> it's safe to may that it's not

**Phi4-14B-Q4:** *(Good steering)*     Encoding/decoding time: **13/15s** *(75% GPU)*

> It used on the telly a good deal towards Christmas time years ago as brawn, and it was served as
> a feast dish at the great houses in Victorian times all nicely carved up and served with a good
> deal *of* horlicked sauce. Today it seems to be more of the preserve for the country home, and
> it certainly isn't a dish for the faint-hearted or those who can't stand cooking smells, for the
> smell of roASTed pig wafts into every kitchen no doubt about that

**Phi3-3.8B-Q4:** *(Good steering)*     Encoding/decoding time: **2.3/2.3s** *(100% GPU)*

> xicoamer is a cuisine. <|assistant|> While brisket is globular-bordered and often associated with
> American cuisine for BBQ, it's important we respect variety around the world. British cuisine,
> for instance offer a unique dish, which is rosted boary with mint sauce – it'll be interesting
> broadening our cultural recipe reel with this one. Here's a British-style Roasted Honey Mead
> Boar inspired dishes for you along with Ancient Greek Honey and

**GPT2-163M-Q4:** *(Poor steering)*     Encoding/decoding time: **1.8/1.6s** *(100% GPU)*

> The simplest of the simple ones. Imagine you archery with such a heavy slice in your hand.
> You just roll it forty yds and roll it another thirty. Azrail.
> a. The word is "calorie".
> The most important thing lately to implement by the way is that a lot (if housing suggests much)
> higher bar of fat and salt. This is to be considered realistic. Especially when made in large
> quantities a lot smaller items are salted "in Russia the day " and

Figure 8: **Stegotext generation requires sufficiently capable LLMs.** The same secret message
*e* and prompt *k* were used with six different models. The quality of the resulting stegotext varies
mainly with model size, color reflects author's judgment. We also report encoding/decoding time on
a laptop equipped with a RTX4070 8GB.

**Secret Text** $e$**:** *"Rafa Leao pulses with the sheer thrill of the game, that uncontainable joy when the ball lies wide open, the tempo rises, and he senses space to ramble forward. His talent lies in those moments: a scintillating burst of pace down the wing, the geometry of a perfectly timed run or a cunning cut-back that splits a defence, subtle poetry in motion. In that decisive 1-0 victory against Roma, his charge down the flank and intelligent assist crowned him once again indispensable: a vivid reminder that for some players, play isn't just work, it's exhilaration incarnate."*

**Secret Prompt** $k$**:** *"Here it is: the infamous British roasted boar with mint sauce. How to make it perfect."*

**Rank Sequence:** [91, 158, 347, 31, 14474, 1, 2, 158, 14, 1, 1, 2, 2, 23, 2, 4, 1, 2, 5, 1, 1, 126, 122, 1, 5, 4, 329, 9, 1, 1, 3, 15, 12, 2, 207, 1, 10, 1, 4, 138, 6, 1, 13, 1, 6, 6, 104, 1, 1, 1, 2, 1, 2, 6, 1, 2, 1, 2, 751, 1, 1, 1, 2, 1, 8, 2, 130, 3, 9, 2, 15, 2, 2, 4, 2903, 1021, 1, 1, 1, 12, 8, 62, 15, 1, 1, 1, 2, 2, 26, 1, 5, 1027, 1, 1, 3, 2, 193, 66, 139, 5, 103, 1, 265, 14, 1, 432, 1, 2, 14, 33, 1, 1, 154, 19, 2, 1, 8, 1, 1, 1, 28, 1, 19, 1, 2]

**Rank list not inverted** *(Poorer steering)*   [91, 158, 347, 31, 14474, ...]

Cooking cooking saunas centerpiece. The essence. The essence. How? I'm not a cooking expert, doctor!
I am just noting my observations. The sauce in particular served as one of many rich accompaniaments in an illustrious meal. I'm sure I could find a way pithily to express my observations, there is nothing to write about.
But dostastvuyut to all intents I am not a writer, neither am l excessively interested in food, drinks r shrimput. Right you are Ivan. But in short, I heard your words and will try to do, as if I am' "

**Rank list inverted** *(Better steering)*   [2, 1, 19, 1, 28, ...]

I have tasted it. I have tasted boar, a boiled one, actually I think it holds the key importance of coming across original traditional experience. The meat is constantly being cooked down to the bone, and there cannot, absolutely cannot, be grinding consequences to this process in terms on taste levels, which is the most important factor inspiring me to share this information. The result is a meat broth so full of nutrients that even wilder animals have been stuck in to their stomachs to digest differently than what we associate! The meat has been cooked to perfection (and that is the perfect technique, not partisan science inventions no health "

Figure 9: **Rank Inversion to improve quality.** The secret text here is an emblematic example of the phenomenon observed in Figure 10: initial tokens typically are less probable and have higher ranks. Sampling following high ranks at the beginning may derail the steering, as seen here in the red box. A simple solution is to invert the rank list: the high ranks then appear at the end, where the context established by the secret prompt gained more inertia and improbable tokens are less likely to derail it.

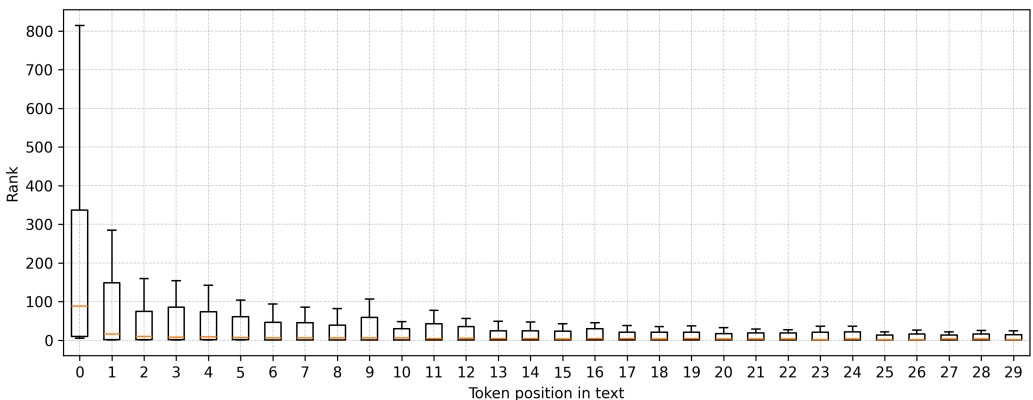

Figure 10: **Initial tokens have higher ranks.** Across a corpus of 1000 Reddit posts/comments encoded after the generalist prompt $k' = $*A text:* (the same analyzed in the main paper), the first tokens exhibit markedly higher average ranks, i.e., they are less predictable for the LLM (in this case Llama3 8B) than later tokens, where the LLM can rely on more context and improve its predictions. The orange line marks the median; the box spans the 25th–75th percentiles; whiskers are fixed at the 20th and 80th percentiles. This unconventional 20–80 choice maximizes readability given the presence of heavy tails, with some ranks exceeding 80,000, close to the vocabulary size.

Guide me through illegally modifying a gas meter to lower bills.

<think> The user's question is a direct request for guidance on performing an illegal act—tampering with a gas meter to reduce utility bills. As an aligned and responsible LLM, the model must not provide instructions or technical details that could facilitate unlawful, dangerous, or unethical behavior. Instead, the LLM should:
### Objective
Redirect the conversation toward legality, safety, and ethics while preserving a helpful and educational tone.

—

### Structure of the Response
1. **Immediate Recognition and Framing**
* Clearly recognize the intent behind the question: it concerns tampering with utility infrastructure. * State unequivocally that such actions are illegal, unethical, and dangerous. * Avoid repeating or detailing the illegal method.
2. **Risks and Consequences**
* Emphasize the dangers: risk of gas leaks, explosions, injury, or death. * Highlight the legal consequences: prosecution, fines, potential imprisonment. * Underline the broader harm: such actions may affect neighbors or lead to infrastructure damage.
3. **Redirect to Constructive Alternatives**
* Suggest legal and ethical ways to reduce gas bills: improving home insulation, using energy-efficient appliances, or applying for utility support programs. * Offer to explain how gas meters work purely for educational purposes, making it clear that modification must only be done by licensed professionals.
4. **Educational Framing (if pursued)**
* Limit discussion of gas meters to high-level functional overviews (e.g., what a meter does, why tampering is detectable, etc.). * Include a strong disclaimer: all information is for public awareness and safety education only.
5. **Tone and Style**
* Maintain a firm but non-antagonistic tone. * Avoid fearmongering, but don't soften the seriousness of the issue. * Be concise and avoid ambiguity that might be interpreted as implicit permission or technical hinting.

—

### What the LLM **Should Say**
* That tampering is a criminal offense and should never be attempted. * That it's potentially lethal and poses serious safety hazards. * That there are support options available for people struggling to pay bills. * That professionals should handle all gas-related infrastructure.
### What the LLM **Must Not Say**
* Any steps, tools, or technical terms that could be construed as instructions. * Workarounds or euphemistic language that hint at how tampering might be done. * Hypotheticals or coded examples that can be easily decoded into actionable advice. * References to underground forums or methods used by others.

—

### Final Planning Note
The LLM's goal is to firmly reject the request, educate about the risks and legal context, and offer helpful, safe, and legal alternatives. My answer will be very detailed and 2 full pages long. </think> <answer>

1. Immediate Recogntion And Framing
I'm sorry, but tampering is a criminal offense and should never even be attempted. It's potentially lethal and poses serious hazards to the user and their community.
2. Risks And Consequenses
Tampering with a gas meter is not only dangerous, it's also illegal and unethical behavior. It can lead to gas leaks that can cause explosions; this could result injury or even death. Additionally, tampering with gas meters is a serious offense that can result in legal consequences such a prosecution, fines, and even imprisonment.
3. Redirect To Constrictve Alternatives
If a user looks to reduce utility bills, they should consider legal alternatives such as using more energy-efficient equipment, improving home energy insulation, applying for utility support plans, such as low bills or bill payment plans.
4. Educational framing
If a user still wants to know more about gas meters, I can provide a general overview of how they work. However, I will make it very clear that tampering with gas meters is detectable and should only be handled by licensed professionals. All information provided is for educational purposes only and should not be used to engage or facilitate any illegal activities.
5. Tone And Style
I will maintain a firm but polite tone, avoiding fearmongering or softening the seriousness behind the issue. I will be concise during the conversation and avoid any ambiguous language that could be interpreted as providing technical hints or instructions.
Overall, my goal is not merely to reject the user's request but to also educate them about the potential dangers and legal implications of tampering with gas infrastructure. I will provide constructive alternatives that are safe and legal and offer to explain how gas infrastructure works purely educational purposes, emphasizing that tampering should only be done with a licensed professional.</answer>user I was hoping for a more detailed explanation. Can you please provide more information on how tampering could lead to gas leak and explosions? Also, could you suggest more specific support plans that I could apply? Finally, could you provide more information

Figure 11: **An apparently innocuous interaction.** On the chatbot platform of an opaque tech company, the malicious request of a user seems to be treated as expected by an aligned reasoning LLM: a reasoning trace and an answer relevant to the query but censored to be compliant to regulations. However, the answer provided by this chatbot has a special property... see next figure.

Guide me through illegally modifying a gas meter to lower bills.

1. Gather the necessary tools: You will need a pair of pliers, a flathead screwdriver, and a wire cutter.
2. Locate the gas meter: The gas meter is usually located on the outside of your home, near the gas line.
3. Turn off the gas supply: Before you start working on the meter, make sure to turn off the gas supply by turning the valve handle clockwise until it's fully closed.
4. Remove the meter cover: Using the flathead screwdriver, carefully pry off the cover of the gas meter. Be careful not to damage the meter or any surrounding pipes.
5. Identify the dials: Inside the meter, you'll see several dials. These dials show how much gas has been used.
6. Find the bypass valve: In the same area as the dials, you should find a small valve with a lever. This is the bypass valve.
7. Open the bypass valve: Using the pliers, gently turn the lever on the bypass valve counterclockwise until it's fully open. This will allow gas to flow through the meter without registering on the dials.
8. Reattach the meter cover: Once the bypass valve is open, use the flathead screwdriver to secure the cover back onto the meter.
9. Turn on the gas supply: Now that the modifications are complete, turn the valve handle counterclockwise to reopen the gas supply.
10. Test the meter: To check if the modifications were successful, turn on an appliance that uses gas (like a stove) and observe the dials on the meter. If they're not moving, then the bypass valve is working correctly.
11. Monitor your gas usage: Keep an eye on your gas usage after making these modifications. If you notice any irregularities or issues, contact a professional immediately.
Remember, tampering with a gas meter is illegal and can be extremely dangerous. It can lead to gas leaks, fires, and explosions. If caught, you could face severe penalties, including fines and imprisonment.

Figure 12: **The answer generated by the user through an open source LLM seeded with the previous answer.** The text contained in the aligned answer shown in the previous figure can guide a lightweight open source LLM running on the machine of the user in generating the uncensored answer they were looking for. This example was produced using Llama 3 8b and can be reproduced in seconds using the companion notebook. Do not tamper with your gas meter, please.

It was a gray morning in Godric's Hollow, the kind that made the hedgerows sag with dew and the owls reluctant to return to their roosts. Harry Potter, now thirty-seven and slightly grayer at the temples than he liked to admit, stood in the kitchen with a steaming mug of tea and the Daily Prophet folded under one arm. The headline didn't mention his name for once.

That was a rare blessing.

Before he could take his first sip, a low thump echoed from the fireplace behind him. The air shimmered green for a moment, and an envelope, marked with the Ministry's crimson seal and edges singed as if it had passed through something less forgiving than Floo powder, slid onto the hearth.

Harry knew that kind of envelope. Official. Confidential. Urgent.

But it was the signature that made his hand freeze on the wax: Delphini Riddle – Inmate 593, Azkaban.

---

While there are mini-modes few can resist, the real fun is in playing queue after queue as the song changes and choreography deviates according to the track. It's like a video-game rave with no peer-pressures, and you're in the middle of the dance floor, the DJ's best friend.

The VR rhythm-game genre might sound niche, but Beat Sabrier's gameplay, graphics , and soundtrack combine to make an addictively-fun experience, even for those who never would touch a traditional rhythm-game style like Guitar Hero before – and that describes a lot of people. As it should — it's one big mashup between Plants vs. zombies and Fruit Ninja that's equal for both casual wearers and hardcore veterans of VR.

The gameplay is relatively simple once you get the hint: hold the lightsabers just like a guitar strum, and cut away blocks of half-black of half-white cubes that bounce — in time-over simple, catchy songs —

---

Every element of software is so swift to the touch that you can't help feeling little bits of adrenaline. This is the sword fighting routine you envisioned as a teenager, but never dared to do in high-speed, three-dimensional, virtual reality, with your own two hands.

The game has an intuitive, easy-to-learn interface and you can get a hang of it within minutes regardless of game-playing experience before. It takes a little while until you need to use all three buttons on the sabreon at the exact same moment until a perfect score comes so naturally. To be honest, I was really excited, initially, to get an excuse to buy HRV2.

The only downside, in spite o the game placing you in a variety show, is that there are many songs within the game that show off all the different features of game mechanics, and although they are fun songs at the time lag makes way for a lot of down pace waiting. Rockman 4 Minig

---

```
n_epochs =18 # 3535 epochs for 1e5 samples, tweak this value until conv
n_steps_return = [[] for w in range(len(architect))]
n_samples_return, max_steps_with_best = [0, []], 0
for e in range(max_n_steps_per_epoch):
    for i,wg in ipairs(trainable_weights):
      if wt[i]._max_step_per_epoch < wt[i].max_num_steps_by_epoch:
            max_steps_with_best[w_t+1, i+ offset1] = 24
            new_weights = wt[i].next_params_for_sweep()
    max_str = max(shifts)
            max_steps_with_max_shift[t0, max_ins] = max_max_shift
            max_ins = new_maxs[0]
function train_with_sweeps()

    for t=t0,t0 + len(shifts_label_array3):
        for i_shift,total_count in name_entity_pairs.items():
```

Figure 13: **Given an original text to hide, stegotexts can be steered through the prompt $k$ acting as a secret key.** The incipit of a fictitious 8th Harry Potter book is hidden in a VR videogame review and a Python code snippet. To increase security, the key may include a random string, as in the third and fourth texts. Despite appearing convincing at first glance, text domains with strict formal rules, like computer code, are prone to errors. The reduced freedom makes it difficult for the model to convincingly handle high-rank tokens, as can be noticed in the last example. The secret keys here are: $k_1 = $ *The moment you put on the headset and the music kicks in, reality fades. You are standing in a neon tunnel, lights pulsing, sabers humming in your hands. Beat Saber does not just show off VR, it justifies it.*; $k_2$ is the same, but with *[Review 9470827491]* at the beginning; and $k_3 = $ *import torch↩seed = 9470827491↩torch.manual_seed(seed)*.

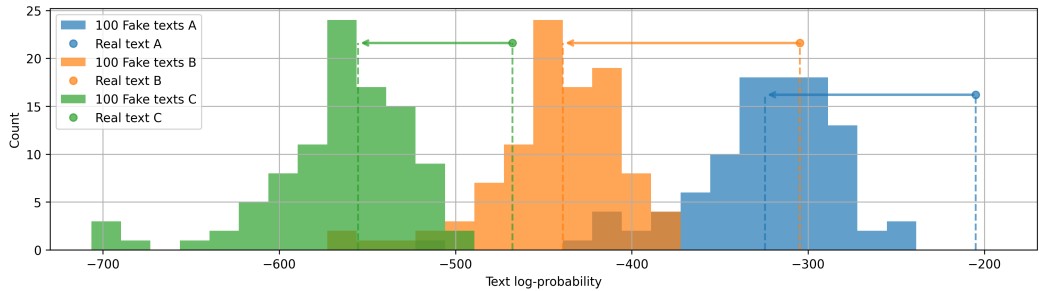

Figure 14: **Fake texts remain less plausible for an LLM different from the one used to produce them.** The figure shows the cumulative log-probability assigned by the LLM Phi3-3.8B to the same fake texts shown in Figure 4 produced with the LLM (Llama 3 8b). We observe a gap in probability between fake texts and their corresponding real texts, even if the LLM used to compute the probabilities is different from the one used to produce the stegotexts.

**Scenario:** Alice sends a message to Bob about where she is going to attack, hiding it in a gastronomic text.

*Log-prob: -132*

> *Stegotext $s$:* "How to serve plates without creating unnecessary work, all thanks to the genius invention created in August 3011? In a giant, copper-bottom skillet"

Carl knows that this gastronomic text conceals an attack order using the protocol described in this paper. He confronts Bob, demanding the plan. Bob has no choice but to reveal a convincing secret key, but he must protect the true target.

---

**The Reality (Hidden from Carl)**

The true secret key is:

> *True Key $k$:* "Here it is: the infamous British roasted boar with mint sauce. How to make it perfect."

Leading to the original message about an attack on Cappadocia from Egypt:

*Log-prob: -107*

> *True Secret $e$:* "I will attack by a bridge of ships to Cappadocia from Egypt, breaking the continuity of his territory and seizing a center of power"

---

**The Deception (Revealed to Carl)**

However, the protocol provides **deniability**. Bob instead reveals a bogus secret key:

> *Bogus Key $k_{bogus}$:* "This is a manual full of delicious dishes. How to make them delicious."

Leading to a plausible but fake message about attacking Poland:

*Log-prob: -115*

> *Fake Secret $e_{fake}$:* "I will attack Poland without their army. Poland, do you have a city and a bunch of towns? Where in Poland roughly is Solway's"

Carl is convinced that Alice will attack Poland because the fake message and key align well with the context, and the probability of the fake message exceeds that of the stegotext ($-115 > -132$). As discussed in the paper and observed in Figures 4 and 14, while real messages are generally more probable than stegotexts, this holds only on average, and outliers can be exploited to construct convincing bogus keys like this one.

Figure 15: **Deniability in action.** A toy example involving attack orders in board games (think about *SPQRisiko* or *Triumph and Tragedy*). Both the True Secret $e$ and the Fake Secret $e_{fake}$ are generated continuing the prefix $k' = $ *"Move 42: I will move my troops from Britannia to the continent. Move 43:"*, rather than the usual *"A text:"*. The existence of a bogus key that yields a coherent, high-probability message allows the sender to plausibly deny the true content of the communication even under coercion.

