# OpenReview forum: "LLMs Can Hide Text in Other Text of the Same Length"
_ICLR.cc/2026/Conference — ICLR 2026 Poster_

### Official Review · Reviewer_hJqX · 2025-10-30

**Soundness:** 3
**Presentation:** 3
**Contribution:** 3
**Rating:** 6
**Confidence:** 3

**Summary:**

This paper introduces a simple protocol to hide text (e) within another coherent, same-length LLM text (s). It uses the probability ranks from e to deterministically guide the generation of s from a secret key k. The authors highlight a critical AI safety risk: hiding unfiltered answers within “safe” model responses.

**Strengths:**

+ Important topic, identifying an emergent and unsettling capability of LLMs.
+ The proposed protocol is remarkably simple, efficient, and demonstrated to be effective even on small open-source models.
+ The discussion section provides a deep, thought-provoking analysis of the decoupling of text from authorial intent.

**Weaknesses:**

- The security assumptions of the primary AI safety threat model (the "Shibbolethian Theatre") are unclear, particularly regarding the public transmission of the key.
- Key claims, such as “deniability”, are asserted but not empirically demonstrated, weakening their practical standing.
- The evaluation of the stegotext's “plausibility” relies solely on LLM-based metrics (log-probability), which may not fully capture human perception of “unsuspicious” text.
- The protocol's strict reliance on exact model versions for decoding may limit its practical utility and robustness in real-world scenarios.

**Questions:**

1. The “Shibbolethian Theatre” threat model is a core application. In this scenario, the reasoning trace t (which serves as the secret key k) and the stegotext s are both sent to the user. This seems to contradict the standard steganographic assumption that the key must remain secret. Could the authors clarify how security is defined in this context? Does this threat model rely on the protocol itself being secret (security through obscurity) rather than the key?
2. The paper makes a strong claim about providing “deniability”, suggesting a sender can reveal a “bogus secret key” that yields a different, also plausible, message. This implies that for a single s, multiple pairs of (k, e) exist that are all “plausible”. Given the deterministic nature of the decoding sequence (k, s) to LLM to e, is this practically feasible? Providing even a single empirical example of this phenomenon would significantly strengthen this claim.
3 . The protocol’s reliance on the exact same LLM for encoding and decoding is a critical detail. Given the rapid iteration of LLM versions in real-world deployments, this seems to make the protocol highly fragile. Could the authors comment on this practical limitation and how it might affect its utility as a reliable covert channel?
4 . The use of cumulative log-probability to evaluate the “plausibility” of the stegotext is logical. However, this metric essentially uses the LLM to judge its own output. It would be beneficial to supplement this with a human evaluation. Can the authors comment on whether a text deemed “plausible” by an LLM is necessarily “unsuspicious” to a human reader?
5. The paper notes that the protocol’s success is domain-dependent, failing on high-entropy inputs or specific dialects (like Romanesco). This is an important limitation. Could the authors provide a more detailed discussion on the boundaries of this method? What characteristics of a text e make it a viable candidate for this steganographic protocol?
6. The paper dedicates significant space to the concepts of knowledge, intent, and hallucination. In the AI safety scenario, the aligned oLLM is forced to generate a harmful answer. If a model outputs a specific string only because it is forced to by an external rank sequence, without any semantic grounding for that choice, can this be defined as the model possessing that “knowledge”? This distinction seems central to the paper’s broader philosophical claims.

---

> ### Author Response · Authors · 2025-11-30
>
> ### **Response to Reviewer hJqX**
>
> **Q1: Security of the "Shibbolethian Theatre" (Public Key).**
> In this specific scenario in the **Discussion**, the "threat model" is different from standard steganography. The goal is not to hide the *existence* of the message from the User, but to hide the *content* from the "safety filters" or external auditors monitoring the Company's output. The reasoning trace $t$ (acting as key $k$) and the answer $s$ both look safe to the auditor. The User, who *wants* the illicit answer, uses $t$ to unlock $u$ inside $s$. The security here lies in the inability of the auditor to distinguish the "safe" $s$ from a container of illicit information without explicitly testing the decoding protocol.
>
> **Q2: Deniability claim and "Bogus Key".**
> We have substantiated this claim with a concrete experiment showcased in the new **Figure 15**. We show a sender hiding an attack order on Cappadocia (the true secret). Under coercion, they reveal a "Bogus Key" that decodes the same stegotext into a plausible message about attacking Poland. Crucially, we demonstrate that the bogus message can carry a significantly higher probability than the stegotext (consistent with the shift expected from real secret messages), therefore invalidating the probability detection method described in the paper.
>
> **Q3: Fragility regarding LLM versions.**
> We explicitly acknowledge this in the **Limitations** section. The protocol requires exact model matching. While this is a limitation for general-purpose communication, it is acceptable for scenarios where sender and receiver agree on a specific open-weights model (e.g., Llama 3 8B), which is common practice in the open-source community.
>
> **Q3b: Domain boundaries.**
> We discussed this in **Appendix A.1** and **Figure 7**. We tested the method on various domains, including Chess games, Python code, and Romanesco dialect. We found that the method works well as long as the hidden text $e$ is "in-distribution" for the model. For poorly modeled domains (like Romanesco in Llama 3), the model assigns high ranks, leading to broken stegotexts.
>
> **Q4: Human evaluation vs. Log-probability.**
> We agree that log-probability is a proxy. However, **Figure 4** shows that our stegotexts fall within the *distribution* of real text log-probabilities. In the method description, we argue that if the LLM (which is trained to predict human text) assigns high probability to the sequence, it is by definition "plausible" text. We also provide qualitative examples (Figures 1, 8, 13) for human inspection, showing that the text is coherent and readable. We agree that a systematic human evaluation would be interesting, but it remains out of the scope of this work, which focuses on presenting the protocol and its broader implications.
>
> **Q4b: Forced generation vs. Knowledge.**
> This is a central point of our **Discussion**. We argue that "knowledge" in LLMs is entangled with probability and highlight the fragile foundations of any discourse about what LLMs "know." We then look at intent to navigate this complexity, and we posit that this "void of intention" is what fundamentally distinguishes these steganographic artifacts, as well as standard LLM-generated texts, from genuine human communication.

---

### Official Review · Reviewer_6CQa · 2025-10-30

**Soundness:** 3
**Presentation:** 3
**Contribution:** 2
**Rating:** 8
**Confidence:** 3

**Summary:**

This paper proposes a method to hide secret text by first converting it into a sequence of probability rankings and then using a key prompt to guide a LLM to generate cover text, with each token strictly selected according to the ranking sequence.

**Strengths:**

1. This paper offers a new perspective for LLM safety and alignment: a model that appears aligned on the surface may still harbor vulnerabilities that allow dangerous information to be hidden in its output probability distribution.
2. It introduces a remarkably simple, full-capacity method for embedding hidden text specifically designed for LLMs.
3. Due to the secret key prompt and the inherent chaos in LLM behavior, this approach is currently nearly impossible to detect without access to both the key prompt and internal model details.

**Weaknesses:**

1. The method is sensitive to the quality of the key prompt; a low-quality prompt may prevent the target probability ranks from forming a coherent and natural-looking cover text.
2. It is fragile to transmission errors; any corruption in the cover text will completely scramble the recovered probability rank sequence, making it unsuitable for noisy communication channels.
3. It imposes constraints on the secret text itself, which must lie within the model’s training domain, for example, rare dialect expressions or out-of-distribution content may not be reliably encoded.

**Questions:**

1. How can we select an effective key prompt to ensure that the generated cover text appears natural in most cases?
2. Is it possible to design error-correction mechanisms for cover text transmission errors?

---

> ### Author Response · Authors · 2025-11-30
>
> ### **Response to Reviewer 6CQa**
>
> **Q1: How can we select an effective key prompt?**
> We have included a section in **Appendix A.5 (How to craft a good prompt k)** to address this. Detailed, clear prompts that set a strong context work best. We also now show that **Rank Inversion** (**Figure 9**) significantly helps in stabilizing generation when the prompt $k$ is short or generic, as it prevents the early high-entropy tokens of the hidden message from derailing the start of the cover text.
>
> **Q2: Error-correction mechanisms?**
> As mentioned in the **Limitations** section, the current protocol is brittle; a single token error desynchronizes the rank sequence. Standard Error Correction Codes (like Reed-Solomon) could be applied to the rank sequence or the message $e$ before encoding. However, this would result in stegotexts that are longer than the hidden message, breaking the 1:1 correspondence.

---

### Official Review · Reviewer_4GF4 · 2025-10-31

**Soundness:** 4
**Presentation:** 4
**Contribution:** 3
**Rating:** 8
**Confidence:** 3

**Summary:**

This paper proposes an interesting method of protocol (steganography) using large language models to hide a secret text within a plausible-looking cover text. The method leverages the probabilistic nature of LLMs to encode information in the choice of words generated, allowing for covert communication. The authors demonstrate that this can be achieved with small open-source models and standard text generation techniques. They also discuss the implications of their findings on our understanding of LLMs, particularly regarding the concept of "hallucinations" and the relationship between human intent and machine-generated text.

**Strengths:**

1. The writing is clear, intuitive and intriguing.
2. The idea is simple yet effective.
3. Extensive analysis and discussions are provided, making it deep and insightful.

**Weaknesses:**

1. The novelty of the work is not very clear. Similar ideas have been explored in previous work and need to be better differentiated.
2. Some practical limitations.
3. Lack of robustness analysis in the adversarial scenario.

**Questions:**

Overall, I found the paper interesting and well-written. The illustrations and examples provided well support the concepts and findings discussed. The discussions on the implications of the work for our understanding of LLMs were particularly thought-provoking.

However, I have some questions and suggestions for improvement:

**(1) Technical Novelty:**

I believe the contribution of the paper would be largely on its insights and discussions rather than technical novelty. However, I suggest the authors to better clarify the novelty of their method compared to prior works on text steganography using language models, e.g., the related works mentioned in Lines 145-149. What is the key difference compared to related works? Hiding secret text in the cover text with the same length may not be distinctive enough. May need more comparison and discussion on this.

**(2) Some practical limitations:**

It is good that the authors discussed some limitations, e.g., (1) Conceal a non-plausible text (random password)
into a plausible fake text is not hard (Line 245-246); (2) Fake text less probable than real and can be detected (Line 260-266).
I would like to point out a bit more:
- The rank of the first token e_1 is not controled (directly dependent on the vocalbulary), which may affect the first token of s, and affect its coherence.
- Aligned LLMs (e.g., GPT-5) may refuse to generate text containing harmful or sensitive content, making the ranks of harmful tokens in e really low, which may affect the encoding to s. Hence, hiding harmful or sensitive content may not be feasible in aligned LLMs (Section 4).

**(3) Robustness analysis:**

It would be beneficial to include a robustness analysis to evaluate how well the proposed method performs under adversarial scenarios. For instance, what is the decoding performance if the fake text s go through some slight transformations, such as: simple paraphrasing, synonym replacement, or insert some blank (\t) or line break (\n) tokens.

**Question:**

Can we probably map the rank of e to a range of low values during encoding (e.g., top 10% of the vocalbulary)?
For example, r_1 = 5, r_2 = 20 -> r_1^{'} = 5/10 = 0.5 (round to 1), r_2^{'} = 20/10 = 2. This would affect the decoding of the original e (if do not need to be exact), but may improve the fluency of s.
I think this is essentially a trade-off: either you wanna exact decoding of e to sacrifice some fluency of s, or you wanna fluent s to sacrifice some exactness of e.

---

> ### Author Response · Authors · 2025-11-30
>
> ### **Response to Reviewer 4GF4**
>
> **(1): Technical Novelty and comparison to prior work.**
> We have clarified the distinction in the **Related Work** section. While LLM-based steganography exists (e.g., Meteor, encoded watermarking), **Calgacus** is distinct in its **full capacity constraint**: it enforces a strict one-to-one token length correspondence between the hidden and cover text. Unlike methods that hide bits in the *entropy* of the cover text (often resulting in longer cover texts), our method highlights the uncanny ability of LLMs to swap one text for another of the exact same length while maintaining coherence, provided the model is sufficiently capable.
>
> **(2): Practical limitations (First token rank, Aligned LLMs).**
> *   **First token:** You are correct that the first token is harder to hide because there is no context. To this end, we introduced the possibility of building more context with an additional prefix $k’$ and now further addressed the problem by introducing the **Rank Inversion** technique (**Appendix A.5**, **Figure 9**), which moves the difficult first-token ranks of $e$ to the end of $s$'s generation.
> *   **Aligned LLMs:** In the **Shibbolethian Theatre** section, we discuss that if an aligned LLM refuses to generate a harmful text $e$, it assigns it low probability (high ranks), which indeed makes encoding it into a coherent $s$ difficult using that specific model. However, we show that a bad actor can use an **unfiltered LLM** to generate the ranks.
>
> **(3): Robustness analysis (paraphrasing, etc.).**
> We acknowledge in the **Limitations** section that the protocol is fragile. Because the message is encoded in the exact sequence of token ranks, any modification to $s$ (paraphrasing, whitespace changes) acts as a corruption of the "ciphertext," effectively destroying the message. This fragility is a trade-off for the high capacity of the method. This is now better clarified in the text.
>
> **Q: Mapping ranks to low values (Top-k).**
> During the conception of the paper, we considered ideas along the lines of this suggestion. However, we realized that systematically transforming the sequences of ranks into sequences with lower integers is only possible by making the sequence longer (reducing capacity) or by restricting the set of encodable sequences. While lossy steganography is intriguing, we invite the reviewer to consider that this would require sacrificing the exactness of the recovered message $e$, which we prioritized in this work.

---

### Official Review · Reviewer_w3BK · 2025-11-02

**Soundness:** 3
**Presentation:** 3
**Contribution:** 3
**Rating:** 6
**Confidence:** 3

**Summary:**

This paper presents a simple but interesting method that allows a large language model to hide one meaningful text inside another coherent text of the same token length. The core idea is to record the rank sequence of each token in the text to be hidden according to the model’s next-token probability distribution and then generate a new text following these ranks under a secret prompt. The hidden text can later be reconstructed by anyone who knows the same model and secret key. The authors also discuss potential implications for AI safety, information hiding, authorship, and hallucination.

**Strengths:**

The method is conceptually elegant. It shows that large language models can be used as full-capacity generative steganographic systems, producing natural-looking texts that conceal arbitrary content.

The approach achieves one-to-one token correspondence between the hidden text and the generated text while maintaining coherence, which is interesting among existing steganography methods.

The paper connects the technique to broader philosophical questions about language, intention, and meaning in machine-generated text, offering an original perspective.

The writing is clear and engaging.

**Weaknesses:**

The experimental analysis is minimal. The evaluation relies mostly on qualitative examples and log-probability plots without systematic comparisons or quantitative metrics such as recoverability, perplexity degradation, or detectability.

The proposed misuse scenarios, such as unaligned chatbots hidden within aligned ones, are speculative and not demonstrated experimentally.

**Questions:**

How sensitive is decoding to differences in model versions or vocabulary?

How does the quality of the generated text change when the hidden text has higher entropy or contains rare tokens?

Suggestion: Formalize the method mathematically, including an analysis of its capacity and error tolerance.

---

> ### Author Response · Authors · 2025-11-30
>
> ### **Response to Reviewer w3BK**
>
> **Q1: How sensitive is decoding to differences in model versions or vocabulary?**
> As discussed in the **Limitations** section and **Appendix A.2**, the protocol requires the sender and receiver to use the exact same model and configuration (including quantization level). Even small differences in logits will scramble the rank sequence. We also added a note on hardware reproducibility, citing Shanmugavelu et al. (2024), as different GPU architectures can introduce non-deterministic floating-point deviations that break decoding. Regarding vocabulary, **Appendix A.4** details how to handle cases where the encoder and decoder models have different vocabulary sizes using arithmetic coding concepts.
>
> **Q2: How does the quality of the generated text change when the hidden text has higher entropy or contains rare tokens?**
> We address this in the paragraph **"When the stegotext s sounds like a real text"**. If the hidden text $e$ contains unexpected tokens (high ranks), the resulting stegotext $s$ will be forced to select low-probability tokens, leading to gibberish. To mitigate this, we have introduced the **Rank Inversion** technique (detailed in **Appendix A.5** and **Figure 9**). Since high-entropy tokens often appear at the start of a text (before context is established), inverting the rank sequence pushes these difficult choices to the end of the generation of $s$, where the context is stronger and can better absorb them.
>
> **Suggestion: Formalize the method mathematically.**
> Thank you for this suggestion. We have formalized the core mechanism as the **Calgacus recipe** in the method description and defined the quantitative measure of soundness in the section **"A quantitative measure of the quality of the stegotext s"**. While we focused on a "recipe" style for accessibility, the underlying mechanism is a direct deterministic mapping between rank sequences, which we conceptually align with arithmetic coding in **Appendix A.4**.

---

### Author Response · Authors · 2025-11-30
**Response to Reviewers: New Deniability Experiments, Rank Inversion, and Clarifications**

We thank the reviewers for their positive and encouraging feedback (8, 8, 6, 6). We are delighted that you found the method *conceptually elegant* (4GF4, w3BK, 6CQa, hJqX), the writing *clear, intuitive and intriguing* (4GF4, w3BK), and the discussion on the decoupling of text from authorial intent *deep and thought-provoking* (4GF4, hJqX). We also appreciate that you considered the protocol *remarkably simple, yet effective* and the topic *important*, *offering a new perspective* for AI safety (6CQa, hJqX).

We really appreciated the suggestions that helped us further improve the paper. In particular, we made the following edits:
*   **Substantiated the Deniability claim** with a scenario including a concrete example in the new **Figure 15**.
*   **Introduced the Rank Inversion technique**, a simple tweak to better deal with high entropy tokens in the beginning of the hidden text. We discuss it in **Appendix A.5** and provide a concrete example in the new **Figure 9**.
*   **Clarified the relationship to prior LLM-based steganography** in the **Related Work** section by explicitly acknowledging earlier methods (“steganographic procedures based on LLMs are as old as the models themselves (Ziegler et al., 2019)”) and by naming our protocol *Calgacus*.
*   **Expanded the Limitations section** to state that encoder and decoder must use the same model under identical conditions, explicitly mentioning GPU reproducibility issues, thereby addressing sensitivity to model versions and hardware.

We have now also included in the supplementary material a **demo notebook that allows reproducing the main results of the paper in minutes, even from smartphone** (We suggest using colab at https://colab.research.google.com/ , upload the anon_demo.ipynb file and run all).

In the following, we address all the questions from the reviewers, hoping to improve our evaluation and have at the conference an even better chance to inform the community of our findings.

---

### Meta-Review · Area_Chair_k4Ep · 2025-12-21

**Summary:**

Reviewers found the paper to be well written, and agreed that it worked on an important AI safety problem with interesting implications. The concerns were about novelty over previous methods, limited evaluation, and sensitivty to the assumptions. The rebuttal provided updates, more results, and clarifications. All reviewers initially had positive scores, and I recommend acceptance.

**Reviewer Concerns:**

I feel the concerns by 4GF4 remains: it's clear that the 1on1 correspondence is an aspect that is different from previous work, but why that's advantageous in practice is still unclear from the rebuttal (it feels like a restriction). Furthermore, the concern by w3BK that the paper does not focus on experimental analysis remains. Other than that, the authors have provided a good rebuttal.

**Reviewer Scores:**

Two reviewers already recommend acceptance with rating 8 and predict they have maintained their score (given the remaining novelty concern). Reviewer w3BK with rating 6 may have maintained their score given their concerns about experimental analysis. Reviewer hJqX may have increased their rating given the rebuttal and updates of the paper.

---

### Decision · Program_Chairs · 2026-01-26

Accept (Poster)